# Experimental Implementation of a Magnetic Levitation System for Laser-Directed Energy Deposition via Powder Feeding Additive Manufacturing Applications

**Parichit Kumar** , **Mazyar Ansari** , **Ehsan Toyserkani** and **Mir Behrad Khamesee** *

Department of Mechanical and Mechatronics Engineering, University of Waterloo, Waterloo, ON N2L 3G1, Canada; p43kumar@uwaterloo.ca (P.K.); m6ansari@uwaterloo.ca (M.A.); etoyserk@uwaterloo.ca (E.T.)
* Correspondence: khamesee@uwaterloo.ca

**Abstract:** Magnetic levitation and additive manufacturing (AM) are two fields of significant interest in academic research. The use of non-contact forces for magnetic levitation techniques provides opportunities for adoption within the AM environment. The key goal of this article is to experimentally validate the implementation of a magnetic levitation system for Laser-Directed Energy Deposition via Powder Feeding (LDED-PF) Additive Manufacturing applications. Through simulations (conducted in ANSYS Maxwell) and experimental implementation, the levitation system's stability is tested under a variety of different conditions. The experimental implementation highlights the feasibility of a magnetic levitation system for LDED-PF applications. The levitation system developed is capable of the suspension of non-magnetic materials. The system is also able to maintain stable levitation for extended periods of time. The incorporation of the levitation system into the AM environment may result in an increased maneuverability of non-clamped structures for AM deposition operations.

**Keywords:** magnetic levitation; metal additive manufacturing; Laser-Directed Energy Deposition via Powder Feeding Additive Manufacturing; eddy current induction





## 1. Introduction

Magnetic levitation is defined as the suspension of an object using only non-contact forces induced through interactions of magnetic fields. Magnetic levitation and manipulation have been used in several applications, such as drug delivery within the human body [1,2], energy harvesting techniques [3,4], and the micromanipulation of microrobots [5] amongst several others.

The critical challenge for magnetic levitation systems is their dependence on the ferromagnetic properties of the materials used. Conventional magnetic levitation techniques are heavily reliant on the use of ferromagnetic materials, such as iron, to facilitate stable suspension for extended periods of time [6–8]. While there has been some research associated with the stable levitation of non-ferromagnetic materials such as aluminum [9,10], there is significant scope for growth for the stable levitation of non-magnetic materials.

Additive manufacturing (AM) is the process of melting and joining materials to make parts from 3D model data, usually layer upon layer, as opposed to subtractive manufacturing and formative manufacturing methodologies [11]. Metal AM entails the use of metallic materials to build a part layer by layer. Metal AM has found significant research interest in fields such as aerospace [12], dentistry [13], and repair and reconditioning applications [14] amongst several others. There are seven different types of AM techniques.

The emphasis for this research is placed on Laser-Directed Energy Deposition via Powder Feeding (LDED-PF), an AM technique also known as Laser Powder-Fed Additive Manufacturing, which is a metal additive manufacturing (AM) process that uses a focused heat source to directly deposit materials as they are fed into the heat source [15]. LDED-PF

was chosen as the key point of emphasis because LDED-PF can be used with a wide range of materials, including metals, alloys, and composites. LDED-PF can also deposit material at high rates, which makes it an efficient and cost-effective manufacturing technique. Finally, LDED-PF can easily integrate a magnetic levitation system, with the levitated geometry serving as the build surface for deposition activities.

AM techniques without magnetic levitation systems are heavily reliant on substrates, which are the build surfaces for fusing layers to form the part [16]. This comes with the requirement of the separation of the part from the substrate, which subsequently results in increased post-manufacturing operations. The use of magnetic levitation techniques facilitates the use of non-contact forces for stable suspension where a substrate can be a part of complex shapes while being floated and tilted using levitation during printing.

The absence of mechanical support with magnetic levitation means there is a reduction in the risk of marks or scratches on the surface of the part being built. This can result in a better surface finish and improved part quality. This results in a reduction in post-manufacturing operations. In conjunction with controlling the nozzle height during deposition, magnetic levitation also provides additional controllability to the system by providing the system with the ability to manipulate the position of the levitated geometry in real time. Finally, the use of levitation techniques can also facilitate both sides of the levitated non-clamped geometry to be used for deposition activities. This results in a significant increase in the available build surface for deposition activities. The ultimate culmination of the incorporation of magnetic levitation techniques in LDED-PF applications would facilitate multi-directional deposition, including deposition from below the levitated geometry.

The research presented in this article aims to highlight a novel magnetic levitation system. The system is capable of facilitating the stable levitation of non-magnetic materials without the incorporation of an active feedback controller. The compatibility of different non-magnetic conductive materials with the magnetic levitation system has been highlighted in [17]. The developed system is then tested within the AM environment. The DMDIC106 machine was used for the testing of the LDED-PF operation. The performance of the levitation system is then compared within the simulation and experimental environment to ensure a reliable and repeatable performance.

## 2. System Description

The system relies on the principle of eddy current induction to produce the lift force necessary for stable suspension. The system is made up of two concurrent coils that carry currents in opposite directions and are embedded within a high-permeability material (referred to as the core). The working principle has been highlighted in previous articles [17,18].

In summary, the time-varying currents through the coils produce time-varying magnetic fields via the principle of Ampere's law. These time-varying magnetic fields induce currents (known as eddy currents) within conductors within the sphere of influence of the source magnetic field using the principle of Faraday's law. Finally, the induced eddy currents interact with the source magnetic field to produce the lift force required to overcome the gravitational force associated with the object being levitated. The geometry suspended is a disc. Different dimensions and materials of the discs are analyzed in this article. Two distinct levitation systems are considered for the analysis. The first highlights the use of laminated sheets to minimize the production of eddy currents within the levitation system core, as highlighted in Figure 1a.

Ref. [19,20] highlights the improvement in the core losses associated with the use of the laminated core. However, the use of laminated sheets in the core reduces the effective magnetic permeability of the core, therefore weakening the magnetic focusing capability and finally reducing the overall performance of the levitation system.

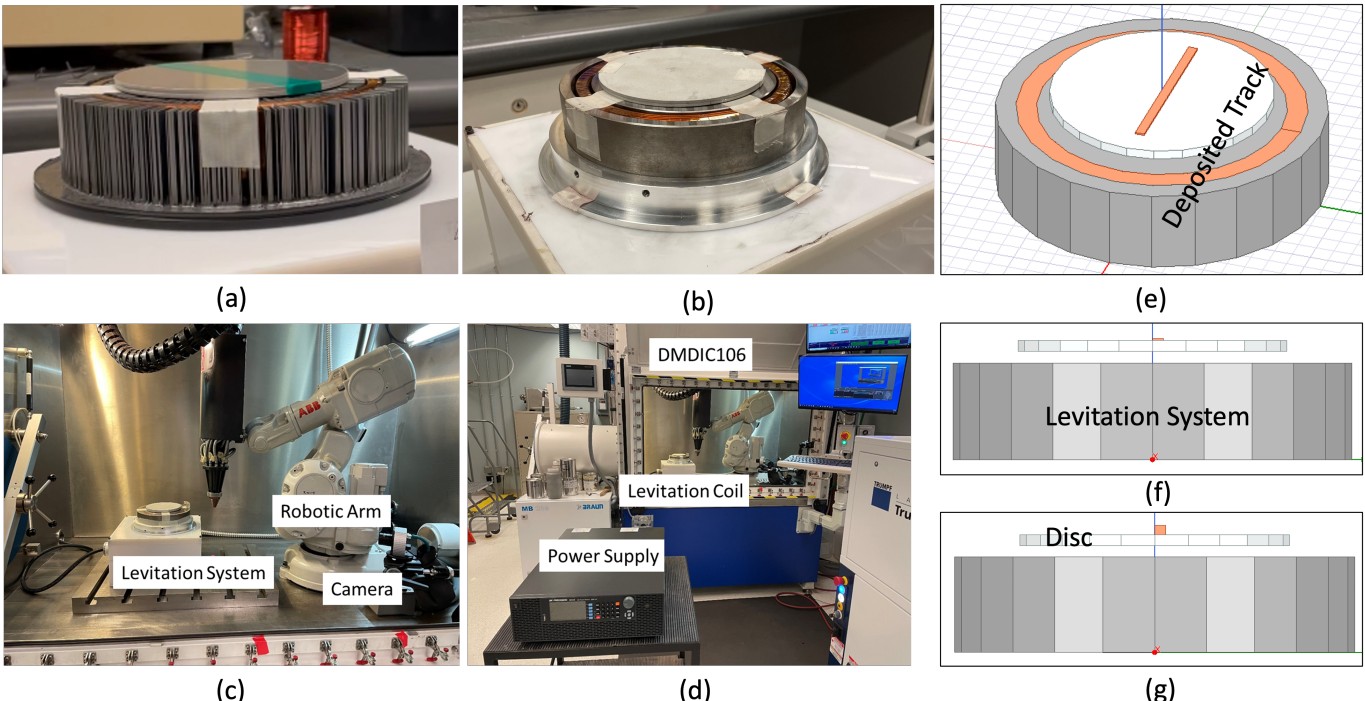

**Figure 1.** (**a**) Laminated core magnetic levitation system. (**b**) Solid core magnetic levitation system. (**c**) Experimental apparatus for magnetic levitation experiment within DMDIC106. (**d**) Experimental apparatus within the DMDIC106 machine. (**e**) Simulation model for ANSYS Maxwell—isometric view. (**f**) Simulation model with only one layer—front view. (**g**) Simulation model with 5 layers—front view

The second levitation system (shown in Figure 1b) considered utilizes a low-carbon steel solid core system with two concurrent coils embedded within it. While this system will produce more core losses due to the induction of eddy currents, the system will maximize its magnetic performance compared to its laminated counterpart. Further details regarding the system schematics, design, simulation, and implementation of the levitation system can be found in [21].

There were several critical considerations that needed to be addressed to facilitate the safe operation of this experimental setup (highlighted in Figure 1c). First, the levitation system's electrical safety was addressed by enclosing all wiring within an enclosure. The enclosure and the levitation system were subsequently covered and sealed to ensure that there is no interaction between the levitation system and conductive intrusive metallic powder dust frequently found within the DMDIC106 machine. Next, the overall volume of the levitation system was only 38.4% of the available working space within the DMDIC106 machine, as shown in Figure 1d. Therefore, sufficient clearance was provided to facilitate all AM operations safely. Finally, the levitated geometry, i.e., the disc, was sand blasted to negate any laser back reflection that might damage the laser. Finally, Figure 1e highlights the simulation model developed in ANSYS Maxwell, a world-renowned software for simulations of electromagnetic systems. The model highlights the levitation system, the levitated disc, and the deposited track. For the analysis conducted in this article, a simple track of 90 mm length and 5 mm width is consistently deposited on the levitated geometry, with each layer resulting in an increase in the height of the deposited track.

### 3. Theory of Impact of Powder Deposition

It is critical to determine the impact forces of powder deposition on the substrate in order to develop a magnetic levitation system suitable for additive manufacturing applications. Since the research presented in this report deals with LDED-PF, the impact forces of powder dispersion are modeled. The analysis presented here makes some assumptions. These are the following:

- The analysis assumes a constant flow of powders and that air friction can be neglected.
- Collisions between particles can be ignored because their sizes are so small and they all come from the same source.

The theorem of transfer of momentum is given by Equation (1).

$$F \triangle t = mv \tag{1}$$

where m is the mass of the powder particle, v is the velocity of the particle, F is the impact force, and $\triangle t$ is the instantaneous time of impact. The impact forces are decomposed to their axial ($F_z$) and radial ($F_r$) directions. This is given by Equation (2).

$$mvsin\theta = F_r \triangle t$$
$$mvcos\theta = (mg + F_z) \triangle t \tag{2}$$

where $\theta$ is the angle of the nozzle and g is the acceleration caused by gravity.

*Skin Depth Effect*

Eddy currents are produced within a characteristic length called the skin depth. There are very minimal eddy currents produced beyond this length since the reactionary magnetic field cancels out the effects of the primary magnetic field. Beyond this length, there are eddy currents induced within the conductor [22]. The skin depth is calculated using Equation (3).

$$\delta = \sqrt{\frac{1}{\pi f \mu \sigma}} \tag{3}$$

where $f$ is the frequency, $\mu$ is the relative permeability, and $\sigma$ is the conductivity of the material.

### 4. Stability of Levitated Disc with Impact of Powder Particles—Simulations vs. Experiments

Table 1 highlights the process parameters selected to conduct the powder deposition operations [23] presents the process parameters for the deposition of copper on a stainless steel disc. It uses a laser power of 600–1800 W, a scan speed of 5–11.6 mm/s, and a powder feed rate of 4–8 g/min. A powder feed rate of 12 g/min was used in this study to verify the stability of the levitated disc with high powder feed rates. The velocity of the powders was calculated using the radius of the outlet of the material nozzle and the volumetric flow rate through the nozzle [24].

**Table 1.** rocess parameters for AM operation.

| Parameter | Value |
|---|---|
| Powder feed rate | 12 g/min |
| Laser power | 1500 W |
| Laser scanning speed | 5 mm/s |
| Powder material | Copper |
| Disc material | A7075, A6061 |
| Velocity of powder | 2.546 m/s |

### 4.1. Simulation Analysis

Having selected the process parameters for AM operations, as highlighted in Section 1, the velocity of the powder particles was the only remaining unknown variable in Equation (2). The calculated powder speed for the material nozzle used in the experiment has been calculated in Table 1. Ref. [25] highlights the powder speeds escaping the nozzle can be as high as 2 m/s. Thus, the calculated velocity of the powder particles is in line with expectations.

To further mimic the 'worst case scenario' analysis, the overall mass deposited after 1 min is assumed to be deposited simultaneously for a small timestep of 0.1 s. To further contribute to the 'worst case scenario' analysis, the overall impact force is assumed to be only in the axial axis.

This ensures that the stability in the axial axis is maintained, despite the extreme conditions of powder deposition. Finally, it is also assumed that the coefficient of restitution of the powder is 0. This implies that the momentum of the power is completely transferred to the levitated disc, therefore producing the maximum impact force. Inputting these values into Equation (2), the resulting impact force is 0.24 N. The stability of the levitation system with this calculated impact of powder dispersion (which is the dispersion of powder on the substrate without a laser to facilitate deposition) for the worst case scenario was tested using ANSYS Maxwell.

The calculated impact force is added as a constant force on the negative x-axis. The levitation force output as a function of time is highlighted in Figure 2a. As highlighted in Figure 2b, the position of the disc stabilizes at 6.75 mm above the levitation system. The resulting data suggest that the levitation system is able to maintain stability even under such extreme conditions. The settling time of this system is 3.67 s. However, through the incorporation of a feedback controller, the settling time can be improved, as discussed in [21].

### 4.2. Experimental Analysis

For the first analysis, the nozzle dispensing the powder was placed at the center of two distinct discs—a 120 mm diameter and 7.5 mm height A7075 disc (Figure 2c) and 120 mm diameter and 5 mm height A6061 disc (Figure 2d). Then, 200 V at 150 Hz (resulting in a 7 A input current) was supplied to the coils in both cases. As highlighted in Figure 2c–h, the discs were able to maintain stability with the addition of powder.

Having established the stability of the levitated disc with a stationary nozzle, it was imperative to highlight the stability of the levitated disc with continuous powder dispersion and a moving nozzle. Thus, an analysis was conducted where the nozzle was moved 90 mm from one side of the disc to the other end. The A7075 disc highlighted in the previous section was used for the analysis. The stability of the disc remained unimpeded by the addition of powder dispersion.

The impact of powder dispersion was also studied at different levitation heights of the disc. This was studied to ensure that the stability would be maintained at inputs lower than the maximum allowable input. It is evident that lower inputs to the levitation coils will not only result in lower axial (z-axis) forces, but they would also result in lower restoration forces in the lateral (x,y) axes. As highlighted in Figure 2e,f, powder dispersion does not impact the overall performance of the levitation system. This in in line with the findings of the simulation analyses conducted in Section 4.1.

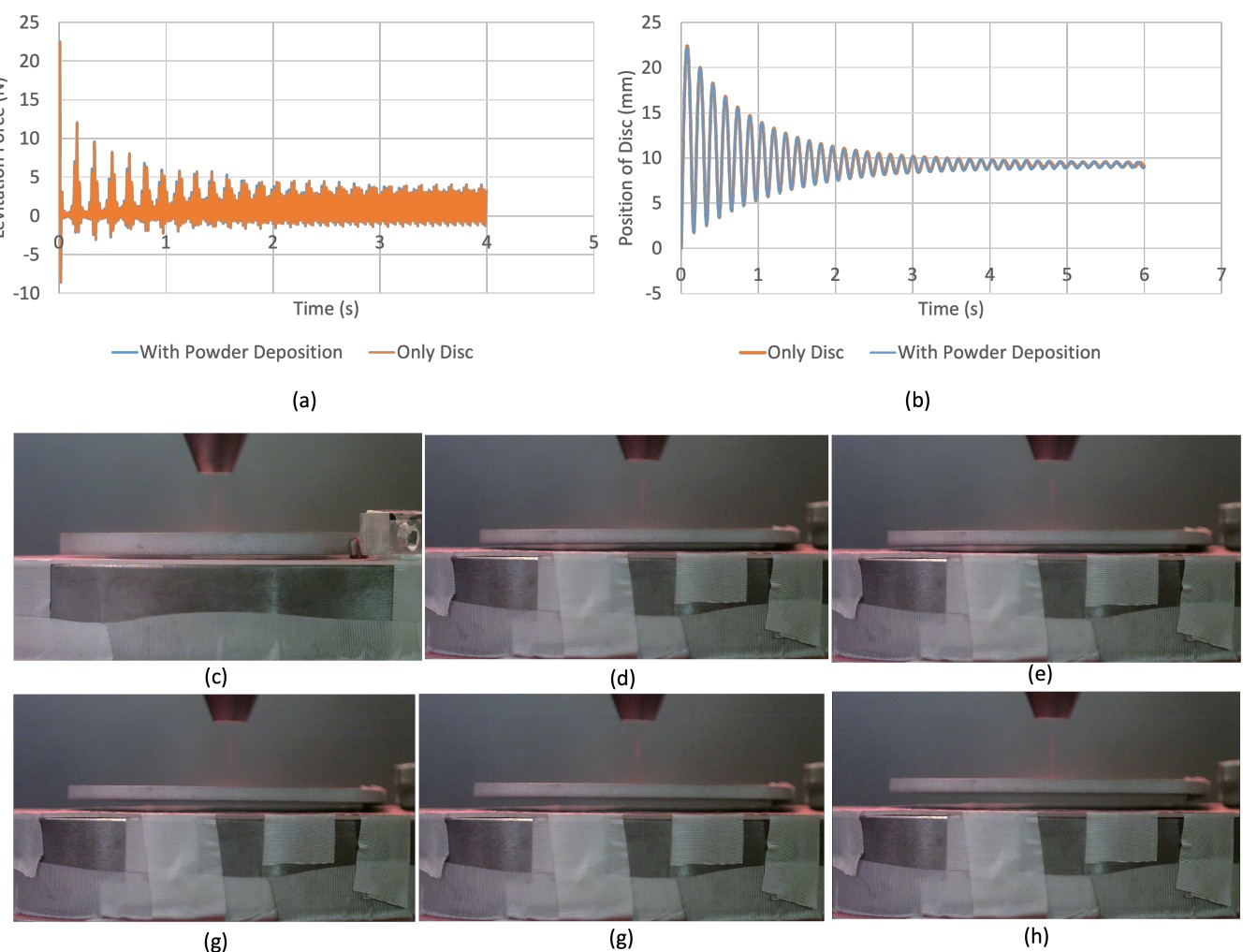

**Figure 2.** (**a**) Levitation force vs. time from ANSYS Maxwell with and without the impact of powder deposition. (**b**) Position of disc vs. time from ANSYS Maxwell with and without the impact of powder deposition. (**c**) Powder dispersion for A7075 disc of 120 mm diameter and 7.5 mm height. (**d**) Powder dispersion for A6061 disc of 120 mm diameter and 5 mm height. (**e**) Powder dispersion for levitation height of 4.9 mm. (**f**) Powder dispersion for levitation height of 6.5 mm. (**g**) Powder dispersion for levitation height of 7.2 mm. (**h**) Powder dispersion for levitation height of 8.1 mm

## 5. Powder Deposition with Laser

### 5.1. Experiment with A7075 Disc

The A7075 disc of 120 mm diameter and 7.5 mm height was utilized for the first analysis. Figure 3 highlights the deposition of copper powder from a layer-by-layer perspective, where 190 V at 150 Hz input (7 A) was supplied to the coils. As highlighted in Figure 3a–f, it is evident that there is an increase in the levitation height of the disc for the same input. Seven layers were deposited successfully.

From Figure 3a–f, the levitation height of the disc can be extracted through simple image processing techniques. Since the height of the disc is known, the gap between the base of the levitated disc and the levitation system can be extracted, using the disc height as a reference. The variation in levitation height with the addition of each layer is highlighted in Figure 4a. The increase in the levitation height can be explained by the additional eddy currents induced within the deposited copper. Copper is a highly conductive material. With the deposition of additional layers, there is an increase in the volume within which eddy currents can be induced.

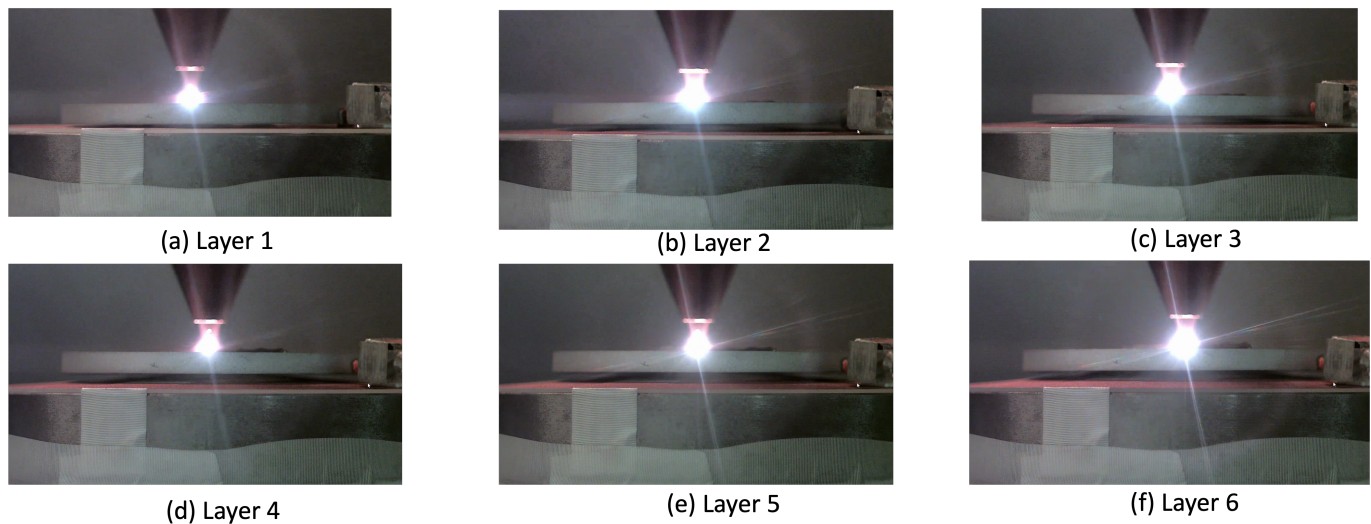

**Figure 3.** Visual representation of layer-by-layer deposition on the levitated geometry

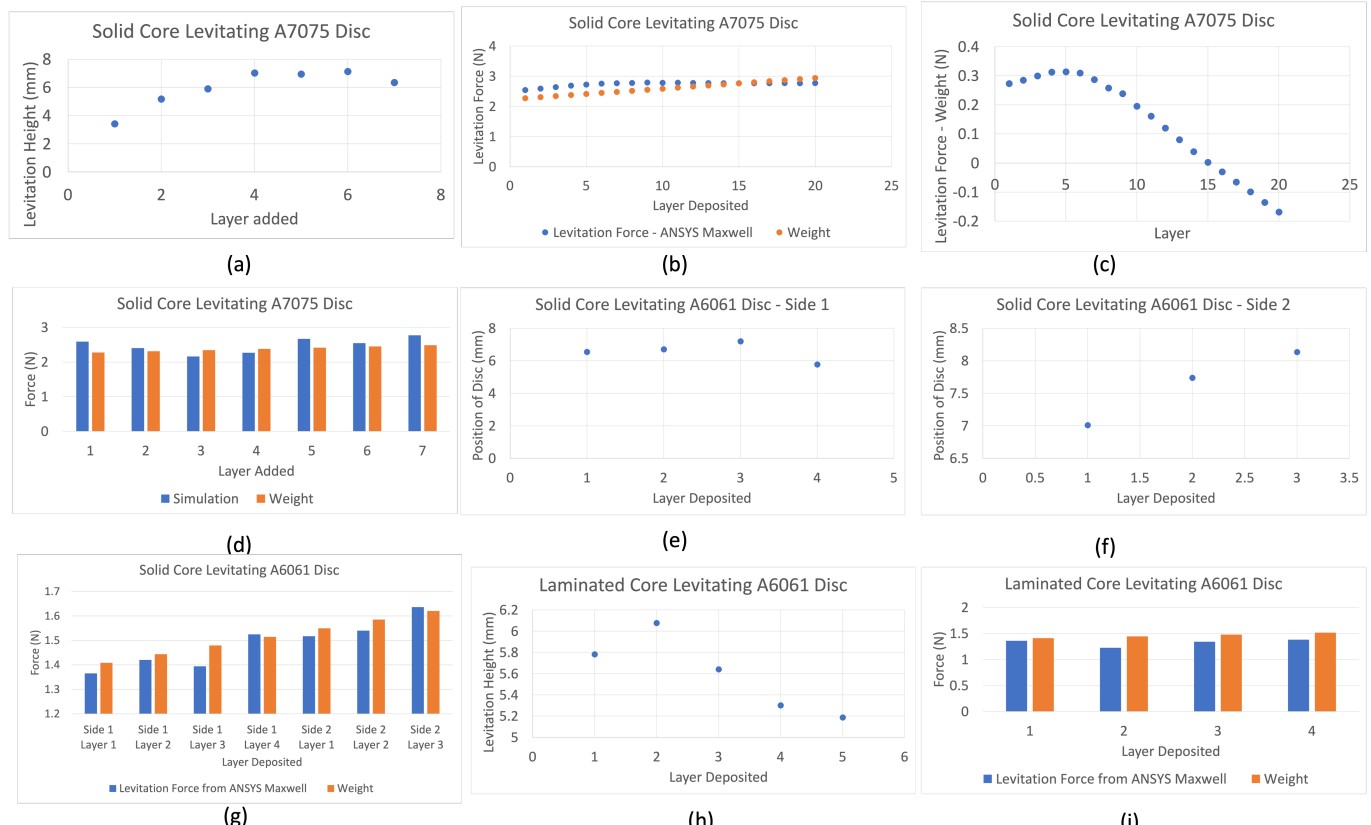

**Figure 4.** (**a**) Layer deposition vs. measured levitation height for A7075 disc. (**b**) Levitation force with increased layers deposited—ANSYS Maxwell. (**c**) Effective force: levitation force—weight from ANSYS Maxwell. (**d**) Levitation force from ANSYS Maxwell and weight vs. layer added at measured levitation height for A7075 disc experiment. (**e**) Layer deposition on side 1 vs. measured levitation height for A6061 disc. (**f**) Layer deposition on side 2 vs. measured levitation height for A6061 disc. (**g**) Levitation force from ANSYS Maxwell and weight vs. layer added at measured levitation height for A6061 disc experiment. (**h**) Layer deposition vs. measured levitation height for laminated core system. (**i**) Levitation force from ANSYS Maxwell and weight vs. layer added at measured levitation height for laminated core system.

This increased volume for eddy current induction results in more eddy current generation, since these deposited layers are still within the skin depth at 150 Hz (the skin depth of aluminum $\delta$ = 6.7 mm and the skin depth of copper $\delta$ = 5.323 mm according to Equation (3)). Thus, there are more eddy currents interacting with the source magnetic field to produce the lifting force responsible for the stable suspension of the aluminum disc. This was tested using ANSYS Maxwell. The layer length, width, and deposited mass per layer were known. The height of each deposited layer was calculated from these parameters. These layers were then fed into ANSYS Maxwell and the levitation force was extracted with the addition of each layer. It should be noted that the disc was held 5 mm above the levitation system for this analysis.

The resulting data of levitation force extracted from ANSYS Maxwell have been highlighted in Figure 4b. As can be seen, there is a clear increase in the levitation force with the addition of each copper layer, confirming the effect of adding a conductor and the subsequent increase in the volume available for eddy current induction. The effective levitation force, which is the difference between the levitation force and the weight of the levitated geometry, was also documented in Figure 4c. As evident, the effective force reaches a peak and then decreases, which is in agreement with the observed trend of an initial rise and the subsequent fall of the levitated geometry with the addition of layers.

The effective force of the levitated disc becomes negative after reaching a threshold beyond which the weight of the disc and deposited layer is greater than the levitation force the geometry is able to produce. This is also in line with expectations. Further testing was conducted using ANSYS Maxwell, where the disc was held at the measured levitation height and the layers were added on top of the A7075 disc. Figure 4d highlights the comparison of the levitation force extracted from ANSYS Maxwell and the weight (i.e., the gravitational force the levitation system must overcome to facilitate stable suspension).

As can be seen, the levitation force extracted from ANSYS Maxwell is in relatively close agreement, with a maximum error of 13% observed. This error can be attributed to the meshing errors of the simulation analysis and slight inaccuracies associated with the measurement of levitation height.

### 5.2. Experiment with A6061 Disc

Following the successful implementation of the first experiment with the A7075 aluminum disc, the analysis was extended to another aluminum alloy—A6061. In addition, the dimension of the disc was also varied to 120 mm in diameter and 5 mm in height. Due to the lower mass of the initial disc, the levitation system would be able to support a higher mass. Another pertinent consideration for the analysis is to maximize the build surface available for AM operation. To facilitate the same, deposition activities were conducted on both sides of the disc. The overall build surface available for AM operations has potentially been increased by facilitating deposition activities on both sides of the levitated disc.

Here, side 1 refers to an arbitrarily selected side used for the first set of deposition activities and side 2 refers to the other side of the levitated geometry used for the second set of deposition activities. As mentioned in the analysis with the A7075 disc, since the height of the disc is known, the levitation height of the disc can be extracted from the images. The levitation height with the deposition of subsequent layers has been highlighted in Figure 4e,f.

The validity of these levitation heights is tested using ANSYS Maxwell. As described previously, the anticipated levitation force at these heights should be close to the weight of the object. The resulting data are plotted in Figure 4g. As can be seen, the levitation force data from ANSYS Maxwell is in relatively close agreement with the experimental data. This analysis successfully highlights the levitation system's ability to levitate a conducting disc (of A6061) and increase the build surface by 100%. The measured levitation height data have been verified through simulation analyses. Thus, the viability of flipping the levitated geometry and maximizing the build surface for LDED-PF operations has been clearly highlighted.

### 5.3. Experiments with Laminated Core System

The laminated core system was also utilized for experimentation within the DMDIC106 machine. A6061 discs of 120 mm diameter and 5 mm height were utilized for this analysis. The levitation coils were given an input of 300 V at 90 Hz, which resulted in 4.35 A of current.

The levitation height was documented using the same techniques presented previously and has been highlighted in Figure 4h. ANSYS Maxwell was utilized to study the levitation force at the measured levitation heights. As highlighted in Figure 4i, the levitation force at the measured levitation height is in close agreement with the anticipated force, i.e., the weight of the levitated geometry.

## 6. Results and Discussions

Figure 4 highlights the measured levitation height and the associated levitation forces at the measured levitation heights with the deposition of copper layers. There is an increase in the levitation height caused by the increase in the eddy current induction. This is in line with [26], since a higher volume of metal results in an increase in eddy current induction.

This results in an increase in the levitation force experienced by the levitated geometry, resulting in the relative rise of the geometry.

This trend is observed until the increase in the weight of the object is higher than the increase in the produced levitation force, following which the levitation height continues to reduce. ANSYS Maxwell confirms the phenomenon. The levitation forces extracted from ANSYS Maxwell for all analyses conducted are within 10% error. These errors can be attributed to the meshing error or slight errors with the measurement of the levitation height.

### 6.1. Successful Levitation and Layer Deposition

The deposition of copper powder on a levitated aluminum disc is successfully presented in this article. Free-formed features built using this magnetic levitation system are shown in Figure 5a. The figure also highlights 3D feature results on both sides of the disc. The deposited layer resulting from the activities presented in Section 5.1 has been presented in Figure 5b.

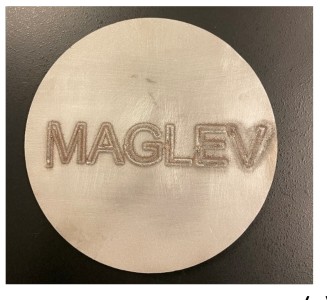 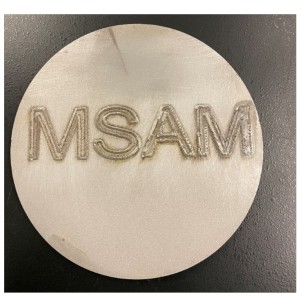

(a)

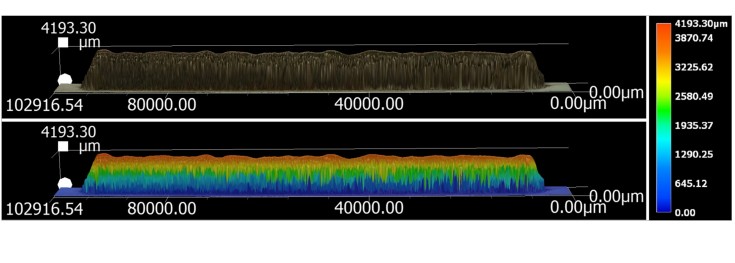

(b)

**Figure 5.** (**a**) Deposition of features on both sides of the levitated geometry. (**b**) Resulting deposited features from the analysis in Section 5.1

The ability of the levitation system to deposit features on both sides of the disc increases the surface area available for deposition activities. Conventional AM operations rely on the use of substrates that are fixed; therefore, they are restricted in the amount of surface area available for material deposition through AM. Through the incorporation of the magnetic levitation system, this constraint can be eliminated.

### 6.2. Effect of Powder Feed Rate

According to [23], conventional copper deposition on stainless steel disc operations requires the use of a powder feed rate of 4 g/min to 8 g/min. Thus, the use of 12 g/min is considered a relatively high powder feed rate for AM operations. In addition, the

simulation analyses highlighted in Section 3 show that the levitation disc retains its stability with a significantly higher powder dispersion force. Thus, the relative stability of the levitation system is clearly highlighted.

Through simulation and experimental analysis, the ability of the magnetic levitation system to support LDED-PF operations has been verified. The magnetic levitation system is able to maintain suspension, despite the incorporation of external forces caused by the impact of powder particles during material deposition.

### 6.3. Effect of Laser Power

According to [23], the laser power utilized for the deposition of copper on stainless steel discs was between 600 W and 1800 W, causing the temperature of the substrate to be high. Thus, the levitation system maintains its stability with a high laser power of 1500 W without losing stability. This highlights the compatibility of the levitation system with a high laser power.

The incorporation of the laser power results in an increase in the temperature of the levitated disc. The magnetic levitation system was able to maintain steady-state stable levitation, despite the increase in temperature, as discussed in more detail in Section 6.4.

### 6.4. Effect of Increase in Temperature of Levitated Disc

Following the deposition of over five layers, there is a steep increase in temperature observed within the levitated geometry, from room temperature (24 °C to 105 °C), measured using a DIGI-SENSE Laser Infrared Thermometer Model 20250-06 before and after the deposition activities. The initial temperature was recorded within the AM machine before the initiation of the deposition experiment and the final temperature was recorded immediately after the deposition experiment.

According to [27], there is a 16% decrease in the conductivity of aluminum with the increase in temperature highlighted. As highlighted in [18], the levitation ability of a material is directly proportional to the conductivity of the material. Thus, a reduction in levitation force is expected. According to the analysis conducted in ANSYS Maxwell, the decrease in conductivity results in a 10% reduction in the overall levitation force experienced by the levitated aluminum alloy disc. However, the system can maintain its stability despite the 10% reduction in the levitation force.

### 6.5. Effect of Input to Levitation System

The input voltage and frequency of the input to the coils within the levitation system play a critical role in the performance of the system. The input voltage and frequency are crucial to determine the levitation height of the levitated disc and the stabilizing restoration forces in the lateral axes to maintain stable levitation over time.

By adjusting the amplitude and frequency of the input voltage, the levitation force can be adjusted and controlled to allow for the adjustment of the levitation height. Thus, it was critical to determine if the levitation system would maintain stability for input lower than the maximum allowable input. As highlighted in Section 4.2, the levitated disc can maintain stability with powder dispersion at different levitation heights.

### 6.6. Effect of Laminated and Solid Cores

Two distinct levitation systems have been presented—a solid core system and a laminated core system. Since the laminated core consists of insulated lamination sheets, the losses caused by the induction of eddy currents have been minimized. However, minimal dimensional flexibility is offered for optimization with a laminated core system, as discussed in [21]. Both systems, however, highlight the ability to produce sufficient levitation forces to facilitate successful LDED-PF operations.

## 7. Conclusions

The present article highlights the first successful implementation of a magnetic levitation system capable of supporting additive manufacturing applications using non-magnetic materials. The primary emphasis is placed on LDED-PF techniques. Two distinct systems were developed for this analysis: one utilizing laminated sheets to build the core and the other utilizing solid low-carbon steel to build the core. The stable suspension of non-magnetic materials, such as aluminum and its alloys, was successfully conducted through experimental implementation.

Next, the levitated disc was also subjected to powder deposition with the laser on. Two different aluminum alloy discs of different dimensions were used for the study. Both systems developed can support the deposition of at least five layers while maintaining stability.

With the addition of a highly conductive material (copper), an increase in the volume of eddy current induction was observed. This subsequently resulted in an increase in levitation force, which produced an increase in the measured levitation height of the disc. The experimental observation was verified through simulation analyses.

The levitation system was subject to different process parameters for the powder deposition of copper on an aluminum disc. With a 12 g/min powder feed rate and 1500 W laser power, the levitated disc maintained stability.

Since the levitated geometry is expected to be a portion of the final built part, there is an anticipated reduction in the number of post-manufacturing operations. The technique has the potential to rotate using magnetic fields with the notion to deposit on all surfaces of the geometry. This tilting idea will be explored in future work. Through the incorporation of a feedback controller, the performance of the levitation system can be improved further.

**Author Contributions:** Magnetic levitation and additive manufacturing: manufacturing, methodology, analysis, investigation, data collection, formal analyses, and preparation of the first draft—P.K. Additive manufacturing: experimental implementation of LDED-PF machine, parameter definition for AM operation, review, and editing—M.A. Additive manufacturing: conceptualization, methodology, supervision, funding acquisition, project administration, review, and editing—E.T. Magnetic levitation: conceptualization, methodology, supervision, funding acquisition, project administration, review, and editing—M.B.K. All authors have read and agreed to the published version of the manuscript.

**Funding:** This work was by the umbrella of the Holistic Innovation in Additive Manufacturing (HI-AM) Network through the Natural Sciences and Engineering Research Council of Canada (NSERC) and Canada Foundation for Innovation (CFI).

**Institutional Review Board Statement:** Not applicable.

**Informed Consent Statement:** Not applicable.

**Data Availability Statement:** Data available from the authors upon request.

**Conflicts of Interest:** The authors declare no conflict of interest.

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
