# Peer review of "Experimental Implementation of a Magnetic Levitation System for Laser-Directed Energy Deposition via Powder Feeding Additive Manufacturing Applications"

_actuators, doi:10.3390/act12060244_

Round 1

Reviewer 1 Report

1.      Article paragraphs need to be shortened.

2.      Note that the equation symbols must be consistent with the font style of the article

3.      The discussion is shallow and needs more details, the observations and future trends. This chapter should be connected with others published papers.

4.      Conclusions should provide a clearer and more concise description of the experimental data.

Author Response

Comment 1: Article paragraphs need to be shortened.

Thank you for your comment. The paragraph lengths have been shortened. Minor edits have also been made to reduce the length of the paragraphs.

Comment 2: Note that the equation symbols must be consistent with the font style of the article

Thank you for your comment. Use of Latex (Overleaf) equation package adhering to MDPI’s guidelines have now been used to ensure consistency of the equation symbols.

Comment 3: The discussion is shallow and needs more details, the observations and future trends. This chapter should be connected with others published papers.

Comment 4: Conclusions should provide a clearer and more concise description of the experimental data.

Thank you for your comment. The discussion and conclusion has been made more concise and a brief discussion of future trends was added as well. Comparisons with 4 articles have been presented in the discussion section to better connect the results of this article with other published paper.

Reviewer 2 Report

The paper describes simulation and experimental realization of a magnetic levitation system for additive manufacturing (AM) using laser-directed energy deposition via power feeding (LDED-PF). To the best of my knowledge, the work is novel and original. The title of the paper is appropriate. The abstract summarizes the approach and the main results. In the introduction, the necessity and relevance of the work is appropriately mentioned and motivated, and it is put into context with other prior works. In the conclusion, the results are briefly summarized. Regarding the intermediate sections, I think they are mostly very well written, however I do have some questions and a few suggestions for further improvement, as listed below.

1)      I suggest to change the title of section 2. Section 2.1 is rather devoted to general design considerations. Only section 2.2 contains theory.

2)      Figure 2a: For early times, it looks like the levitation force oscillates with about 6 Hz frequency. Beyond 2 s, there is a broad blue band developing? Can you explain the origin of this observation? Is there an underlying higher oscillation frequency? A better resolution could help in analyzing this effect. I suggest to rescale the x axis, showing just the first 4 seconds of data. Alternatively, spectral analysis of the time trace by FFT could help resolving that question. In addition, it can be seen that it takes a few seconds until (almost) equilibrium sets in. Is there a way to reduce that time by introducing more damping? Or is the delayed equilibration acceptable?

3)      Figure 4: I suggest writing in each graph the respective core (“A7075”, “A6061” or “laminated”) as a courtesy for the reader, because it is somewhat uncomfortable to extract that from the caption.

4)      Typo in Figure 4f, x axis caption: “Layer”

5)      Typos in caption 4e and 4f: “on side”

6)      Lines 233-234: “To facilitate the same <i.e., maximizing the build surface available for AM operation >, deposition activities were conducted on both sides of the disc.” I do not understand this statement. Please explain. Why did you perform deposition on both sides?

7)      Lines 246-248: “Thus, the viability of flipping the levitated geometry and maximizing the build surface for LDED-PF operations has been clearly highlighted.” This sentence is also unclear to me. Do you mean “flipping the levitated disc”?

8)      Section 5 on Results and Discussion: I missed a discussion of the pros and cons of the three different core system, the solid A7075 and A6061 cores and the laminated core. I suggest to add a subsection here, with a comparison of the results of the three different cores. In addition, I sometimes missed information on which core was used to obtain a certain result.

9)      Lines 259-261: with reference to Figure 4, you state that “There is an increase in the levitation height caused by the increase in the eddy current induction.” Figures 4a and 4f for solid A7075 and A6061 cores show that, but Figure 4h for laminated core shows the opposite, a decreasing levitation height with increasing number of deposited layers. Please formulate the statements more precise, referring to the respective cores, and please discuss this difference.

10)  Lines 313-314: “The levitation system highlights the stable suspension of non-magnetic materials like aluminum and its alloys successfully.” I am not happy with this sentence because it is not the levitation system that highlights something, but rather “The experimental implementation highlights the feasibility of …”, as you write in the abstract.

Author Response

Comment 1: I suggest to change the title of section 2. Section 2.1 is rather devoted to general design considerations. Only section 2.2 contains theory.

Thank you for your feedback. As per your recommendation, the label ‘Theory’ has been removed from Section 2. Section 2 now entails ‘System Description’ and the subsequent section is now titled ‘Theory of Impact of Powder Deposition’.

Comment 2: Figure 2a: For early times, it looks like the levitation force oscillates with about 6 Hz frequency. Beyond 2 s, there is a broad blue band developing? Can you explain the origin of this observation? Is there an underlying higher oscillation frequency? A better resolution could help in analyzing this effect. I suggest to rescale the x axis, showing just the first 4 seconds of data. Alternatively, spectral analysis of the time trace by FFT could help resolving that question. In addition, it can be seen that it takes a few seconds until (almost) equilibrium sets in. Is there a way to reduce that time by introducing more damping? Or is the delayed equilibration acceptable?

      Thank you for your comment. As recommended, Fig. 2 has been updated to present the variation of the levitation force and position of disc over 4 seconds.

      The authors have also developed a separate article where the incorporation of a feedback controller results in the improved performance of the levitation system. Thus, this has been incorporated in the article in Section 4.1, as highlighted below.

“The settling time of this system is quite high. However, through the incorporation of a feedback controller, the settling time can be improved, as discussed in [28].”

Comment 3: Figure 4: I suggest writing in each graph the respective core (“A7075”, “A6061” or “laminated”) as a courtesy for the reader, because it is somewhat uncomfortable to extract that from the caption.

      Thank you for your comment. A title highlighting the levitation system and the material of the levitated disc has been added to each graph.

Comment 4: Typo in Figure 4f, x axis caption: “Layer”

      Thank you for your comment. The error has been fixed.

Comment 5: Typos in caption 4e and 4f: “on side”

      Thank you for your comment. The error has been fixed.

Comment 6: Lines 233-234: “To facilitate the same <i.e., maximizing the build surface available for AM operation >, deposition activities were conducted on both sides of the disc.” I do not understand this statement. Please explain. Why did you perform deposition on both sides?

Thank you for your comment. Conventional LDED-PF operations rely on substrates that can constrained. Thus, the build available for AM deposition activities is restricted to one side. By flipping the levitated substrate, the build surface available for AM deposition activities is increased. Thus, the overall surface area for AM deposition was higher. That is why deposition activities were performed on both sides of the levitated substrate.

A further elaboration of the same has been added in Section 5.2 as follows:

“By facilitating deposition activities on both sides of the levitated disc, the overall build surface available for AM operations has been increased.“

Comment 7: Lines 246-248: “Thus, the viability of flipping the levitated geometry and maximizing the build surface for LDED-PF operations has been clearly highlighted.” This sentence is also unclear to me. Do you mean “flipping the levitated disc”? 

      Thank you for your comment. This is an extension of the explanation offered for the previous comment. Through conducting deposition activities on both sides of the levitated disc, the overall build surface area has been increased.

Comment 8: Section 5 on Results and Discussion: I missed a discussion of the pros and cons of the three different core system, the solid A7075 and A6061 cores and the laminated core. I suggest to add a subsection here, with a comparison of the results of the three different cores. In addition, I sometimes missed information on which core was used to obtain a certain result. 

      Thank you for your feedback. A subsection has been added briefly discussing the pros and cons of the laminated and solid core system. This is highlighted in Section 6.6 and is highlighted in yellow.

“Two distinct levitation systems have been presented - A solid core system and a laminated core system. Since the laminated core is comprised of insulated lamination sheets, the losses caused by the induction of eddy currents have been minimized. On the other hand, minimal dimensional flexibility is offered for optimization with a laminated core system, as discussed in [29]. Both systems, however, highlight the ability to product sufficient levitation forces to facilitate successful LDED-PF operations.”

Comment 9: Lines 259-261: with reference to Figure 4, you state that “There is an increase in the levitation height caused by the increase in the eddy current induction.” Figures 4a and 4f for solid A7075 and A6061 cores show that, but Figure 4h for laminated core shows the opposite, a decreasing levitation height with increasing number of deposited layers. Please formulate the statements more precise, referring to the respective cores, and please discuss this difference.

      Thank you for your comment. The trend of increase in levitation height is observed until the increase in weight of the object is higher than the increase in levitation force caused by the induced eddy currents within the deposited layer. This is evident even within the laminated core system. The levitation height is increased with the deposition of the two layers. However, upon the deposition of the third layer, the increase in weight is higher than the increase in levitation force, which subsequently results in a decrease in the levitation height.

      The explanation has been added in Section 6 as follows and is highlighted in yellow.

      “Fig. 4 highlights the measured levitation height and the associated levitation forces at the measured levitation heights with the deposition of copper layers. There is an increase in the levitation height caused by the increase in the eddy current induction. This results in an increase in the levitation force experienced by the levitated geometry, resulting in the relative rise of the geometry. This trend is observed until the increase in the weight of the object is higher than the increase in the produced levitation force, following which the levitation height continues to reduce.”

Comment 10: Lines 313-314: “The levitation system highlights the stable suspension of non-magnetic materials like aluminum and its alloys successfully.” I am not happy with this sentence because it is not the levitation system that highlights something, but rather “The experimental implementation highlights the feasibility of …”, as you write in the abstract.

      Thank you for your comment. The statement has been adjusted to be in-line with our claim in the abstract. This has been presented below and is highlighted in yellow in the text.

            “The stable suspension of non-magnetic materials like aluminum and its alloys is successfully conducted through experimental implementation.”

Reviewer 3 Report

The article is interesting and the work is correctly describe. Please find below a set of comments willing to improve the article.

1- One main concern. According to the authors, “AM techniques are heavily reliant on substrates” and “This comes with the requirement of the separation of the part from the substrate, which subsequently results in increased post-manufacturing operations.” “The use of magnetic levitation techniques facilitates the use of non-contact forces for stable suspension where a substrate can be a part of complex shapes. The absence of mechanical support with magnetic levitation means there is a reduction in risk of marks or scratches on the surface of the part being built.”

I totally agree on this hypothesis but my concern is how this could be used in real cases. All experiments shown are done using a specific shape (disc) and material (aluminium) for the susbtrate. You must demonstrate that the concept is also feasible for different materials (not two aluminion alloys) and for different substrate shapes that shall be part of the complex 3D final part. Aluminium has been adequately selected as it is very conductinve, but maybe you can do some simulations with steel or titanium as they are materials widely used in metal AM. Also you can choose using square, triangular, smaller or irregular shapes to demonstrate the concept of substrate that can remain in the final part.

2- Please add a diagram or some schematics about the levitation systems shown in Fig  1a and Fig 1b. I’d like to get more details on the design. The pictures and the description, as they are not,  would hardly permit the reproduction of the experiments.

3- Please give more details about the transient simulations: boundary conditions, loads, timesteps, number of elements, computation time. The design of the coils must be given in detail.

4- What happens if the powder is deposited near to the edges of the substrate? Does it tilt?

Author Response

Comment 1: One main concern. According to the authors, “AM techniques are heavily reliant on substrates” and “This comes with the requirement of the separation of the part from the substrate, which subsequently results in increased post-manufacturing operations.” “The use of magnetic levitation techniques facilitates the use of non-contact forces for stable suspension where a substrate can be a part of complex shapes. The absence of mechanical support with magnetic levitation means there is a reduction in risk of marks or scratches on the surface of the part being built.”. I totally agree on this hypothesis but my concern is how this could be used in real cases. All experiments shown are done using a specific shape (disc) and material (aluminium) for the susbtrate. You must demonstrate that the concept is also feasible for different materials (not two aluminion alloys) and for different substrate shapes that shall be part of the complex 3D final part. Aluminium has been adequately selected as it is very conductinve, but maybe you can do some simulations with steel or titanium as they are materials widely used in metal AM. Also you can choose using square, triangular, smaller or irregular shapes to demonstrate the concept of substrate that can remain in the final part.

Thank you for your feedback. The authors have considered the effect of using different conductive materials (including titanium and steel) in another article. This has been referenced in Section 1, as highlighted below and in yellow in the text.

“The compatibility of different non-magnetic conductive materials with the magnetic levitation system has been highlighted in [20]”.

Comment 2 & 3:

Please add a diagram or some schematics about the levitation systems shown in Fig  1a and Fig 1b. I’d like to get more details on the design. The pictures and the description, as they are not,  would hardly permit the reproduction of the experiments.

Please give more details about the transient simulations: boundary conditions, loads, timesteps, number of elements, computation time. The design of the coils must be given in detail.

Thank you for your comments. The authors have published an article highlighting the design, development, and implementation of the magnetic levitation system. This has been referenced in the article in Section 2 as shown below and is highlighted in yellow in the manuscript.

“Further details regarding the system schematics, design, simulation, and implementation of the levitation system can be found in [20].”

Comment 4: What happens if the powder is deposited near to the edges of the substrate? Does it tilt?

Thank you for your comment. As highlighted in Section 4.2, the stability of the levitated disc is maintained when powder dispersion occurs from one side of the disc to the other. The powder deposition of the resulting ‘MSAM’ and ‘MAGLEV’ highlighted in Fig. 5 also possess an offset. However, the levitated disc maintains its stability.

Round 2

Reviewer 1 Report

1.      Do not have too many paragraphs in each chapter or section

2.      The discussion is shallow and more details are needed to explain the substantive significance of the obtained results.

Author Response

The authors appreciate the reviewer’s comments and their thorough review. The changes are made within the revised manuscript and highlighted in yellow. The responses to the reviewer’s comments/questions along with the changes made in the manuscript are noted below.

1.    Do not have too many paragraphs in each chapter or section

Thank you for your feedback. The number of paragraphs presented in each section has been reduced from 7 papers per section (an average) to 4 paragraphs per section. Edits have been made to improve the overall readability of the paper as well.

2.    The discussion is shallow and more details are needed to explain the substantive significance of the obtained results.

Thank you for your comment. Additional details regarding the impact of the result and its associated contribution to the improvements to the Laser Directed Energy Deposition via Powder Feeding process through the incorporation of a magnetic levitation system have been presented more clearly. This has been highlighted below and in yellow within the attached manuscript.

‘6.1. Successful Levitation and Layer Deposition
The ability of the levitation system to deposit features on both sides of the disc increases the surface area available for deposition activities. Conventional AM operations rely on the use of substrates that are fixed, therefore, restricted in the amount of surface area available for material deposition through AM. Through the incorporation of the magnetic levitation system, this constraint has been bypassed. ‘

‘6.2. Effect of Powder Feed Rate

Through simulation and experimental analysis, the ability of the magnetic levitation system to support LDED-PF operations has been verified. The magnetic levitation system is able to maintain suspension, despite the incorporation of external forces caused by the impact of powder particles during material deposition.’

‘6.3. Effect of Laser Power

The incorporation of the laser power results in an increase in the temperature of the levitated disc. The magnetic levitation system was able to maintain steady-state stable levitation, despite the increase in temperature, as discussed in more detail in Section 6.4.’

‘6.5. Effect of Input to Levitation System

By adjusting the amplitude and frequency of the input voltage, the levitation force can be adjusted and controlled to allow for the adjustment of the levitation height. Thus, it was critical to determine if the levitation system would maintain stability for input lower than the maximum allowable input. As highlighted in Section 4.2, the levitated disc can maintain stability with powder dispersion at different levitation heights.’

Reviewer 3 Report

ok

Author Response

Thank you once again for your invaluable feedback. The authors greatly appreciate it. 

Round 3

Reviewer 1 Report

No.